# Learn More with Less: Uncertainty Consistency Guided Query Selection for RLVR

**Hao Yi**[1,2], **Yulan Hu**[2]*, **Xin Li**[2], **Sheng Ouyang**[1,2], **Lizhong Ding**[3], **Yong Liu**[1]*

[1] Renmin University of China, Gaoling School of Artificial Intelligence, Beijing
[2] Amap, Alibaba Group
[3] Independent Researcher
`yihao@ruc.edu.cn`

## Abstract

Large Language Models (LLMs) have recently improved mathematical reasoning through Reinforcement Learning with Verifiable Reward (RLVR). However, existing RLVR algorithms require large query budgets, making annotation costly. We investigate whether fewer but more informative queries can yield similar or superior performance, introducing active learning (AL) into RLVR. We identify that classic AL sampling strategies fail to outperform random selection in this setting, due to ignoring **objective uncertainty** when only selecting by subjective uncertainty. This work proposes an **uncertainty consistency** metric to evaluate how well subjective uncertainty aligns with objective uncertainty. In the offline setting, this alignment is measured using the Point-Biserial Correlation Coefficient (PBC). For online training, because of limited sampling and dynamically shifting output distributions, PBC estimation is difficult. Therefore, we introduce a new online variant, computed from normalized advantage and subjective uncertainty. Theoretically, we prove that the online variant is strictly negatively correlated with offline PBC and supports better sample selection. Experiments show our method consistently outperforms random and classic AL baselines, achieving full-dataset performance while training on only 30% of the data, effectively reducing the cost of RLVR for reasoning tasks.[1]

## 1 Introduction

Large Language Models (LLMs) (Guo et al., 2025; Team et al., 2023; Achiam et al., 2023; Yang et al., 2024) have recently advanced complex mathematical reasoning. A key driver is Reinforcement Learning with Verifiable Reward (RLVR) (Shao et al., 2024), which leverages explicit, verifiable rewards in math tasks (correct vs. incorrect). This property allows us to obviate the need for learned reward models and critics commonly used in actor-critic methods. Instead, group-based normalized rewards are used to estimate sequence-level advantages, as implemented in methods such as GRPO (Shao et al., 2024), DAPO (Yu et al., 2025), REINFORCE++ (Hu, 2025), and RLOO (Ahmadian et al., 2024).

However, these algorithms typically require tens of thousands of queries to reach optimal performance, while annotating answers for mathematical reasoning tasks incurs substantial cost. Moreover, previous RL research has overlooked the influence of query selection on reasoning ability. Poor query selection can bias models, induce entropy collapse, gradient instability, and hinder convergence (Yu et al., 2025; Cui et al., 2025; Cheng et al., 2025; Wang et al., 2025). So this leads to a pivotal question: *Can we achieve comparable or superior performance with fewer but more informative queries in the RL reasoning task?* Active learning (AL) (Wang & Shang, 2014) offers a promising approach.

In classic AL, a model selects a budgeted subset of unlabeled queries for annotation based on uncertainty (Citovsky et al., 2021; Wang & Shang, 2014; Geifman & El-Yaniv, 2017) or feature-space

---

*Corresponding author.
[1]The code is available at https://github.com/yihao-123/uncertainty-consistency.

coverage(Ash et al., 2021; Agarwal et al., 2020; Sener & Savarese, 2017). The goal is to label the most valuable examples, thus improving generalization with minimal annotations. Similar ideas appear in LLMs for in-context learning (Diao et al., 2023; Du et al., 2024), preference alignment (Chen et al., 2024; Muldrew et al., 2024), supervised fine-tuning (Bayer, 2025), and knowledge distillation(Zhang et al., 2023). Therefore, we assess classic AL strategies in the offline reasoning RL setting. We first conduct a pilot experiment with Qwen2.5-0.5B (Qwen et al., 2025) on the MATH dataset (Hendrycks et al., 2021), as shown in Table 1. The results clearly show that classic AL strategies fail to improve performance in our setting. This raises an important question: *why do traditional AL methods fail here?* To investigate this, we perform an additional analysis focusing on the relationship between subjective uncertainty (e.g., perplexity estimated by the model) and objective uncertainty (i.e., rule-based reward), shown in Figure 1a. Consistency samples (orange curve in Figure 1a) are those which subjective and objective uncertainty are simultaneously high or low. However, Figure 1a reveals that samples with high subjective uncertainty but correct answers produce extreme policy gradients. These outlier gradients induce high variance, destabilizing RL training and hindering reasoning performance. In contrast, consistency samples yield more stable training dynamics. For a concrete illustration why inconsistent samples tend to induce larger variance in gradient norms, consider that in a consistent positive sample, the average generation probability of tokens might be around 0.9, whereas in an inconsistent positive sample it may be around 0.3. Since policy gradient methods tend to increase the probabilities of positive sample tokens toward 1, inconsistent samples have a larger margin for probability improvement, thereby making large-norm and unstable gradients more likely. The same reasoning applies to negative samples. Motivated by these findings, we propose that focusing on consistency samples rather than simply those with the highest subjective uncertainty would be more beneficial for stable and effective RL training.

In the offline setting, we measure the uncertainty consistency using the Point-Biserial Correlation Coefficient ($r_{pb}$) (MacCallum, 2002), and select the top $p\%$ samples with minimal $r_{pb}$ for RL. However, estimating $r_{pb}$ becomes challenging in online settings due to limited sampling and dynamical change in output distributions (Guo et al., 2025). To address this, we propose an online uncertainty consistency metric, $r_{pb}^{online}$, calculated from the normalized advantage and the current model's uncertainty. Theoretically, we prove that the online metric is strictly negatively correlated with its offline counterpart, and under mild conditions, its maximization matches optimizing sample-wise uncertainty. These results provide a solid theoretical foundation for the proposed method.

Experimental results show that in the offline setting, selecting samples with lower offline consistency metrics noteblely outperforms random selection and classic active learning strategies. In the online setting, our method achieves performance comparable to or better than RL with the full dataset by training on only 30% of the data. This validates our metrics' efficacy in query selection for RL reasoning.

Table 1: Results of different AL strategies on MATH task using GRPO on Qwen2.5-0.5B. Full denotes the use of the full dataset, while other strategies employ offline sampling with a sampling ratio of 10%. Every experiment is conduct 5 times using different random seed. The result shows that classic AL strategies, whether based on uncertainty (PPL, Entropy (Wang & Shang, 2014)), features (K-means[2], K-center (Sener & Savarese, 2017)), or LLM-based prompting (AskLLM (Sachdeva et al., 2024)), demonstrate **no marked difference** compared to the Random strategy. AskLLM prompt is shown in Appendix C and more experimental setup refers to Appendix D.

| Q2.5-0.5B | Full | Random | PPL | Entropy | K-center | K-means | AskLLM |
|---|---|---|---|---|---|---|---|
| MATH | **33.2** ($\pm$0.1) | 31.0 ($\pm$0.2) | 30.8 ($\pm$0.6) | 29.9 ($\pm$0.3) | 31.1 ($\pm$0.3) | 30.4 ($\pm$0.5) | 30.9 ($\pm$0.7) |

## 2 RELATED WORK

**Reinforcement learning base on verifiable reward (RLVR).** Building on the noteble performance of reinforcement learning methods in aligning LLMs with human feedback, for example

---

[2]Featured by Qwen3-8B-Embedding (Zhang et al., 2025). Choosing the points to be labeled as the centers of k-means.

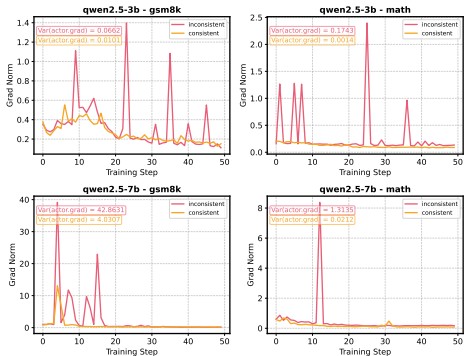
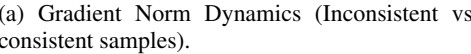

(a) Gradient Norm Dynamics (Inconsistent vs consistent samples).

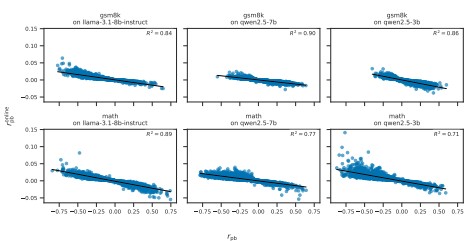

(b) Correlation between $r_{pb}^{online}$ and $r_{pb}$ ($\gamma = 1$). Each point corresponds to one training instance. $K = 64$ responses are sampled for that instance and used to estimate $r_{pb}$ and $r_{pb}^{online}$.

Figure 1: (a) Gradient norm dynamics for inconsistent vs. consistent samples. (b) Correlation between online and offline uncertainty consistency metrics.

PPO (Schulman et al., 2017), Shao et al. (2024) extends Reinforcement Learning base on Verifiable Reward (RLVR) to improve LLM mathematical reasoning capabilities. GRPO (Shao et al., 2024) dispenses with reward model and replaces it with rule-based rewards. It also removes the trainable critic model used to estimate sequence returns and advantages. Instead, it estimates policy gradients using group-wise normalized advantages, which greatly reduces GPU memory usage during RL training. RLOO (Ahmadian et al., 2024) estimates the advantage with a leave-one-out method, improving sample efficiency. REINFORCE++ (Hu, 2025) follows PPO by incorporating KL regularization into the advantage estimate, mitigating reward hacking issues that can arise with the GRPO group-wise estimation. DAPO (Yu et al., 2025) improves sample efficiency and reduces reward noise through empirical techniques such as clip higher, dynamic sampling, and overlong reward shaping. However, these algorithms typically require tens of thousands of queries to reach optimal performance, while annotating answers for mathematical reasoning tasks incurs substantial cost. Moreover, previous RL research has overlooked the influence of query selection on reasoning ability. Poor query selection can bias models, induce entropy collapse, gradient instability, and hinder training (Yu et al., 2025; Cui et al., 2025; Cheng et al., 2025; Wang et al., 2025).

**Active Learning (AL).** The core idea of Active Learning (AL) is: we can maximize post-training performance with a small and limited labeling cost by selecting the most informative queries. Classic AL strategies fall into two categories: uncertainty-based methods and feature-space methods. Uncertainty-based methods assume that the samples with the highest uncertainty are the most informative. Representative approaches include Least Confidence (LC), Margin Sampling (MS), and Maximum Entropy (Wang & Shang, 2014; Geifman & El-Yaniv, 2017). However, these methods ignore the distribution of samples in feature space, so the selected subset may lack diversity. To address this, Core-Set (Sener & Savarese, 2017) and CDAL (Agarwal et al., 2020) expand the coverage of the selected subset in feature space. This increases diversity while preserving the overall information content of datasets. Although Kaplan et al. (2020) shows that LLMs are data-hungry during pretraining and benefit from more data, additional data can hinder optimization and limit achievable performance in many downstream tasks. In in-context learning (ICL), Activate Prompting (Diao et al., 2023) highlights inefficiencies in example selection: the most effective examples are not directly discoverable. CAL (Du et al., 2024) argues that datasets exhibit biased diversity and over-optimization. In supervised fine-tuning (SFT), ActiveLLM (Bayer, 2025) finds that the cold-start phase requires large amounts of data, which limits its utility. In preference alignment tasks, Chen et al. (2024) argues that most current online algorithms still rely on human preference labels provided as feedback to update the policy model. This reliance leads to substantial expert query costs. In knowledge distillation (KD), Zhang et al. (2023) shows that LLMs are easily influenced by erroneous signals, and the high cost of annotation restricts their applicability in domain-specific tasks. However, the effectiveness of AL strategies in the RL reasoning task is underexplored.

## 3 PRELIMINARY

In this section, we first present the unified mathematical formulation of the RLVR loss in Section 3.1. In Section 3.2, we introduce the method for estimating subjective uncertainty.

### 3.1 REINFORCEMENT LEARNING BASED ON VERIFIABLE REWARDS (RLVR)

RL methods epitomized by PPO (Schulman et al., 2017) already serve in InstructGPT (Ouyang et al., 2022) to align LLMs with human feedback. To eliminate the memory overhead of training a critic and a reward model, Shao et al. (2024) introduces RLVR to improve the reasoning capabilities of LLMs. In the on-policy setting, RLVR can unify into the following form:

$$\mathcal{L}_{\text{RLVR}}(\theta|x^{(i)}) = -\frac{1}{K}\sum_{k=1}^{K}\frac{1}{|y_k^{(i)}|}\sum_{t=1}^{|y_k^{(i)}|}\hat{A}_{k,t}^{(i)}\log\pi_\theta(y_{k,t}^{(i)}|x^{(i)}), \tag{1}$$

where $K$ denotes the sample number for each query, $\pi_\theta$ represents the policy model, $|y_k^{(i)}|$ is the length of the $k^{th}$ response and $\hat{A}_{k,t}$ is the token-level advantage function. Usually, the response level advantage function is adopted. For GRPO as an example, $\hat{A}_k = \hat{A}_{k,t} = \frac{R(y_k|x)-mean(\{R(y_k|x)|k\in[K]\})}{std(\{R(y_k|x)|k\in[K]\})}$, $R$ is the reward function. In mathematical reasoning tasks, a rule-based reward function is typically adopted:

$$R(c,a|x) = \begin{cases}1, & a = a^* \\ 0, & \text{otherwise}\end{cases}. \tag{2}$$

Here, $c$, $a$, and $a^*$ represent the reasoning content, the predicted answer, and the reference answer, respectively.

### 3.2 UNCERTAINTY ESTIMATION

To estimate the subjective uncertainty of LLMs, we first use the reference model $\pi_{ref}$ to sample each sample $x^{(i)}$ for $K$ times, obtaining $K$ responses, denoted as $\{y_j^{(i)}\}_{j=1}^K$. For each response, we compute the PPL as follows:

$$\text{PPL}_k^{(i)} = e^{-\frac{1}{|y_k^{(i)}|}\sum_{t=0}^{|y_k^{(i)}|}\log\pi_{ref}(y_{k,t}^{(i)}|x^{(i)},y_{k,<t}^{(i)})}. \tag{3}$$

A larger value of $\text{PPL}_k^{(i)}$ indicates that the probability of $\pi_{ref}$ sampling $y_k^{(i)}$ is lower, which corresponds to a higher degree of subjective uncertainty for the sample $x$. More uncertainty estimation method (Margin Score, Entropy) can refer to Appendix B. For convenience in subsequent discussions, we use $U_k^{(i)}$ to uniformly denote one of the above metrics.

## 4 METHOD

In this section, we introduce the query selection method based on uncertainty consistency under two scenarios, offline and online. Section 4.1 describes how to characterize uncertainty consistency in the offline setting; in Section 4.2, we extend the concept of uncertainty consistency to the online setting, and demonstrate the connection between the offline and online metrics as well as how the online metric affects the single-step optimization process theoretically.

### 4.1 UNCERTAINTY CONSISTENCY IN OFFLINE SCENARIOS

In Section 1 , we have shown that, in mathematical reasoning tasks within RL scenarios, simply applying classic AL strategies, such as selecting samples where the LLM exhibits the highest subjective uncertainty for training, does not lead to better enhancement of reasoning ability compared to randomly choosing training samples. We argue that the key reason for this ineffectiveness lies in the neglect of the relationship between objective and subjective uncertainty.

| Offline Query Selection |
|---|
| **Input:** Query set $\mathcal{D} = \{x^{(i)}\}_{i=1}^N$, reference model $\pi_{\text{ref}}$, offline generations $K$, sample ratio $p$, training steps $T$, mini-batch size $B$, online generations $G$ |
| **Output:** Policy $\pi_{\theta_T}$ |
| 1  Initialize $\theta_0$; |
| 2  **for** $i = 1$ **to** $N$ **do** |
| 3    Generate $K$ responses $\{y_k^{(i)}\}_{k=1}^K \sim \pi_{\text{ref}}(\cdot|x^{(i)})$; |
| 4    Compute $r_{\text{pb}}^{(i)}$ via Formula 4; |
| 5  Select the $p\%$ queries with smallest $r_{\text{pb}}^{(i)}$ to form $\hat{\mathcal{D}}$; |
| 6  **for** $t = 0$ **to** $T - 1$ **do** |
| 7    Sample $\{x^{(j)}\}_{j=1}^B \subseteq \hat{\mathcal{D}}$; |
| 8    **for** $j = 1$ **to** $B$ **do** |
| 9      Generate $G$ responses $\{y_g^{(j)}\}_{g=1}^G \sim \pi_{\theta_t}(\cdot|x^{(j)})$; |
| 10    Compute $\nabla_\theta \mathcal{L}$ via Formula 1; |
| 11    Update $\theta_{t+1} \leftarrow \theta_t - \eta \nabla_\theta \mathcal{L}$; |
| 12  **return** $\pi_{\theta_T}$; |

| Online Query Selection |
|---|
| **Input:** Query set $\mathcal{D} = \{x^{(i)}\}_{i=1}^N$, reference model $\pi_{\text{ref}}$, sample ratio $p$, training steps $T$, mini-batch size $B$, online generations $G$ |
| **Output:** Policy $\pi_{\theta_T}$ |
| 1  Initialize $\theta_0$ from $\pi_{\text{ref}}$; |
| 2  **for** $t = 1$ **to** $T$ **do** |
| 3    Sample mini-batch $\mathcal{B} = \{x^{(j)}\}_{j=1}^B \subseteq \mathcal{D}$; |
| 4    **foreach** $x^{(j)} \in \mathcal{B}$ **do** |
| 5      Generate $G$ responses $\{y_g^{(j)}\}_{g=1}^G \sim \pi_{\theta_{t-1}}(\cdot|x^{(j)})$; |
| 6      Compute $r_{\text{pb}}^{\text{online}}(x^{(j)})$ via Formula 5; |
| 7    Select top-$p\%$ queries with largest $r_{\text{pb}}^{\text{online}}$ to form $\hat{\mathcal{B}}$; |
| 8    Compute $\nabla_\theta \mathcal{L}$ via Formula 1 on $\hat{\mathcal{B}}$; |
| 9    Update $\theta_t \leftarrow \theta_{t-1} - \eta \nabla_\theta \mathcal{L}$; |
| 10  **return** $\pi_{\theta_T}$; |

Figure 2: Offline (left) and online (right) query selection procedures.

In Equation 2, this Bernoulli variable reflects the degree of objective uncertainty: a lower accuracy rate over $K$ samples indicates higher objective uncertainty. Through our experiments, we found that samples with higher alignment between subjective and objective uncertainty tend to be more valuable for the RL reasoning task. Since the reward is a binary variable, for a training sample $x$, we use the PBC to characterize the relationship between subjective uncertainty $U$ (Equation 3) and objective uncertainty $R$ (Equation 2):

$$r_{pb}(x^{(i)}; \{y_j^{(i)}\}_{j=1}^K) = \frac{\bar{U}_1 - \bar{U}_0}{s_K}\sqrt{\frac{K_0 K_1}{K^2}}. \tag{4}$$

Here, $\bar{U}_1$ denotes the mean subjective uncertainty for correct responses for $x^{(i)}$, $\bar{U}_0$ for incorrect responses, $s_K$ is the standard deviation of $\{U_k^{(i)}\}_{k=1}^K$, and $K_1$ and $K_0$ are the numbers of correct and incorrect responses, respectively. A more negative value of $r_{pb}$ indicates a stronger negative correlation between the two variables, and vice versa. Therefore, when subjective and objective uncertainty are aligned, $U$ and $R$ should exhibit a negative correlation, i.e., $r_{pb}$ should be as small as possible. In the offline setting, we pre-select the top $p\%$ of samples with the smallest $r_{pb}$ values from the training set as the dataset for RL. The detailed procedure is shown in Figure 2 (left).

## 4.2 UNCERTAINTY CONSISTENCY IN ONLINE SCENARIOS

In the previous section, we propose an offline uncertainty consistency metric (Equation 4). However, this metric presents the following challenges in online RL settings:

- **Estimation accuracy.** Because the calculation of $r_{pb}$ relies on a large number of samples $K$, it is not feasible to estimate this value accurately in practice;

- **Model update**. The calculation of subjective uncertainty $U$ depends on probability outputs of the model, and in online settings, the sampling distribution of the behavioral policy is changed as the model is updated.

Therefore, we propose an equivalent online uncertainty consistency metric that can mitigate the above issues:

$$r_{pb}^{online}(x^{(i)}; \{y_j^{(i)}\}_{k=1}^K) = \frac{1}{K}\left(\sum_{A_j>0}\frac{\hat{A}_j}{U_j^\theta} + \gamma\sum_{A_j<0}\frac{\hat{A}_j}{U_j^\theta}\right). \tag{5}$$

Here, $\hat{A}_j$ is the advantage estimate assigned by the RL algorithm (e.g., GRPO) for the sample $(x^{(i)}, y_j^{(i)})$, $U_j^\theta$ is the current model $\pi_\theta$'s estimation of the uncertainty metric (see Equation 3) and $\gamma > 0$ is a hyperparameter to balance the ratio between positive response and negative responses.

Based on our experiments (Figure 1b) and theoretical analysis Theorem 1, we find the negative correlation between $r_{pb}$ and $r_{pb}^{online}$. Figure 1b shows the relationship between the offline metric $r_{pb}$ and the online metric $r_{pb}^{online}(\gamma{=}1)$. There is a noteble negative correlation between $r_{pb}^{online}$ and $r_{pb}$ in six combinations of models and datasets. Theorem 1 shows that the covariance between $r_{pb}$ and $r_{pb}^{online}$ is strictly negative. Therefore, to select the most valuable training sample, we choose the top $p\%$ of samples with the largest $r_{pb}^{online}$ from the minibatch at every step during the online training, which means the highest subjective and objective uncertainty consistency. The detailed procedure refers to Figure 2 (right).

**Theorem 1** (Negative Correlation between $r_{pb}$ and $r_{pb}^{online}$). *For the same model $\pi_\theta$, the covariance between $r_{pb}$ and $r_{pb}^{online}$ is less than zero, i.e.,* $\mathrm{Cov}(r_{pb}, r_{pb}^{online}) < 0$.

The proof is shown in Appendix A.1.1. Furthermore, to explain why the largest $r_{pb}^{online}$ samples are effective in the RL reasoning task, we have proven Theorem 2:

**Theorem 2** (Equivalent between Maximizing Decrease in Sample Uncertainty and Maximizing $r_{pb}^{online}$). *Suppose $U(x; \theta) = \sum_{j=1}^K U_j^\theta$ denotes the subjective uncertainty for sample $x$. Under sample gradient orthogonality assumption (Assumption 1) and bounded gradient norm assumption (Assumption 2), in one optimization step of an on-policy RL algorithm (e.g., GRPO), selecting samples in the minibatch with largest $r_{pb}^{online}$ can maximize the decrease in sample uncertainty .*

**Assumption 1** (Sample Gradient Orthogonality). *For any $i, j \in [K]$ and $i \neq j$, the derivatives of the sample uncertainties with respect to the parameter $\theta$ are orthogonal, namely $< \nabla_\theta U_i^\theta, \nabla_\theta U_j^\theta >= 0$.*

**Assumption 2** (Bounded Gradient Norm). *For any $i \in [K]$, we have $0 < m < ||\nabla_\theta U_i^\theta||_2 < M$.*

The explanation of the assumption and the proof is shown in Appendix A.1.2.

## 5 EXPERIMENT

### 5.1 SETUP

**Model**. We select the following models to evaluate the effectiveness of uncertainty consistency metric in both offline and online settings. These models cover different sizes, different architectures, and include both Pretrained and Instruct versions: 1) Qwen2.5-7B (Q-7B) (Qwen et al., 2025) 2) Qwen2.5-3B (Q-3B) (Qwen et al., 2025) 3) Llama-3.1-8B-Instruct (L-8B-I) (Dubey et al., 2024).

**Dataset**. We choose two mathematical reasoning tasks, MATH and GSM8K. MATH consists of 7,500 training examples and 5,000 test examples, while GSM8K consists of 7,474 training examples and 1,319 test examples. All evaluations focus on generalization within each task, without considering cross-task generalization. In order to utilize the CoT ability, we use CoT prompt shown in Appendix C.

**Baseline & Metric**. In the offline setting, we include the following baselines for comparison: 1) **Full**: Using the entire training dataset. 2) **Random**: Randomly selecting p% of the data. 3) **PPL & ENT**: For each query, generating $K = 64$ inferences, then selecting the top p% of examples based on the highest PPL (Equation 3) or ENT (Equation 7) scores. 4) **K-center** (Sener & Savarese, 2017): Utilizing Qwen3-8B-Embedding (Zhang et al., 2025) to extract language features from training queries, and then applying the K-center algorithm to select p% of the data. 5) **AskLLM** (Sachdeva

et al., 2024): The ask prompt we used for LLM is shown in Appendix C. 6) **Active Prompt** (Diao et al., 2023) 7) **CAL** (Du et al., 2024).

In the online setting, Random, PPL, and ENT are performed within each minibatch. All other baselines remain the same as in the offline setting. All experiments report the greedy decoding Pass@1 score on the test set.

**RL Algorithm Hyperparameter.** We mainly use GRPO to validate the effectiveness of our method. Training is conducted with sampling temperature set to 1.0, maximum response length of 2048, and $K = 8$. $\gamma$ in Equation 5 is selected from the set $\{0.1, 0.5, 1.0, 1.5, 2\}$. AdamW (Loshchilov & Hutter, 2017) is used as the optimizer with a constant learning rate of 1e-6, and k3 KL divergence regularization with a coefficient of 0.001. The batch size is set to 256, and total training steps are fixed at 50. In Section 5.3, we further discuss the effects of different RLVR algorithms. The hyperparameters for RLOO and REINFORCE++ are identical to those of GRPO. As for DAPO, we set the high clip ratio and low clip ratio to 0.28 and 0.20, respectively, with a long buffer length of 1024 and a long penalty coefficient of 1.0; Other hyperparameters are the same as GRPO. All experiments are run on $8 \times 96$GB NVIDIA H20 GPUs.

## 5.2 MAIN RESULT

Table 2: Main result in offline and online setting.

| Model | Method | GSM8K | | MATH | |
|---|---|---|---|---|---|
| | | OFF(p=30) | ON(p=30) | OFF(p=30) | ON(p=30) |
| Q-7B | Full | 91.5 | **91.5** | 73.2 | **73.2** |
| | Random | 88.6 | 88.1 | 70.8 | 68.2 |
| | PPL | 88.9 | 90.4 | 71.0 | 72.1 |
| | ENT | 88.4 | 90.3 | 70.3 | 71.8 |
| | K-center | 88.1 | - | 70.5 | - |
| | AskLLM | 87.8 | - | 69.8 | - |
| | Active Prompt | 85.2 | - | 65.1 | - |
| | ZS-CAL | 84.3 | - | 64.0 | - |
| | $r_{pb}$(**Ours**) | 90.1(+1.5%) | - | 72.1(+1.3%) | - |
| | $r_{pb}^{online}$(**Ours**) | - | **91.7**(+2.4%) | - | 72.9(+4.7%) |
| Q-3B | Full | 85.2 | **85.2** | 63.8 | 63.8 |
| | Random | 82.4 | 81.2 | 62.2 | 58.8 |
| | PPL | 81.6 | 83.4 | 61.6 | 62.5 |
| | ENT | 82.7 | 83.2 | 62.9 | 62.8 |
| | K-center | 83.0 | - | 62.5 | - |
| | AskLLM | 82.8 | - | 61.9 | - |
| | Active Prompt | 80.5 | - | 54.1 | - |
| | ZS-CAL | 79.2 | - | 54.5 | - |
| | $r_{pb}$(**Ours**) | 83.6(+1.2%) | - | 63.3(+1.1%) | - |
| | $r_{pb}^{online}$(**Ours**) | - | 84.9(+2.5%) | - | **64.0**(+5.2%) |
| L-8B-I | Full | 90.2 | **90.2** | 52.0 | 52.0 |
| | Random | 87.0 | 88.0 | 50.9 | 50.6 |
| | PPL | 86.5 | 90.5 | 49.8 | 51.6 |
| | ENT | 87.0 | 88.7 | 50.7 | 51.3 |
| | K-center | 87.3 | - | 51.0 | - |
| | AskLLM | 86.2 | - | 50.5 | - |
| | Active Prompt | 84.1 | - | 49.5 | - |
| | ZS-CAL | 84.5 | - | 48.6 | - |
| | $r_{pb}$(**Ours**) | 88.7(+1.7%) | - | 51.5 (+0.6%) | - |
| | $r_{pb}^{online}$(**Ours**) | - | 89.9 (+2.9%) | - | **52.5** (+1.9%) |

The main experimental results are reported in Table 2. We focus on three types of models and conduct comparative studies on two mathematical datasets under both online and offline settings. In all experiments, the sampling ratio is consistently set to 30%.

**Offline.** In the offline setting, classic uncertainty metrics (PPL, ENT), feature-based methods (K-center), and the prompt-based uncertainty method (AskLLM) perform similarly to random selection and are noteblely worse than using the full dataset. These findings are consistent with our preliminary experiments in the warm-up stage. Applying AL strategies in ICL provides only limited improvements to model reasoning ability. In contrast, directly selecting the 30% of samples with the lowest $r_{pb}$ yields much better performance than random selection, although it still falls short of the results obtained with the full dataset. This gap can be attributed to estimation bias caused by the changes in model output distribution during RL parameter updates.

**Online.** In the online setting, selecting samples with high PPL and ENT values during RL leads to better reasoning performance than random selection, but does not reach the performance achieved with Full. However, dynamically selecting the top 30% samples with the highest $r_{pb}^{online}$ in each minibatch not only noteblely outperforms random selection, but also closely matches the performance of Full. In some cases, such as Qwen2.5-7B on GSM8K and Qwen2.5-3B on MATH, it even surpasses Full by 0.2%. These results indicate that in RL-based reasoning training, using only 30% of the data is sufficient to achieve nearly the same performance as Full. Samples with consistent subjective and objective uncertainty are more valuable, supporting both our empirical observations and theoretical hypotheses.

### 5.3 ABLATION STUDY

To further evaluate the validity and robustness of our method, we conducted additional experiments using Qwen2.5-7B.

**Choosing Top p% or Bottom p% $r_{pb}$ samples.** To investigate whether samples with consistent subjective and objective uncertainty are more valuable for RL reasoning training, we selected the bottom 30% (consistency samples, lowest $r_{pb}$) and top 30% (inconsistency samples, highest $r_{pb}$) in the offline setting. We then evaluate model performance on the test set throughout training, as shown in Figures 3 (1-a) & (1-b). The results indicate that, for both MATH and GSM8K, using consistency samples consistently outperforms using inconsistency samples. Moreover, training with inconsistency samples even performs worse than randomly selecting the same number of samples. This suggests that inconsistency samples may be detrimental to model reasoning in RL.

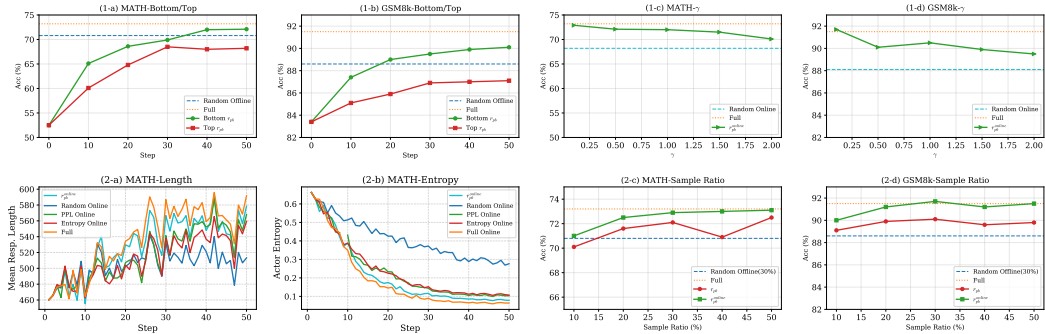

Figure 3: Ablation study and discussion on Qwen2.5-7B: **(1-a)&(1-b)** Choosing Top 30% or Bottom 30% $r_{pb}$ samples. **(1-c)&(1-d)** Sensitivity of $\gamma$ in Equation 5. **(2-a)&(2-b)** Responses length and entropy during online training. **(2-c)&(2-d)** Different Sample Ratios.

**Sensitivity of $\gamma$ in Equation 5.** We explored the impact of different $\gamma$ values, ranging from $\{0.1, 0.5, 1.0, 1.5, 2.0\}$, as illustrated in Figures 3 (1-c) & (1-d). The results show that our approach favors smaller $\gamma$ values, such as 0.1. With larger $\gamma$, performance is lower and does not match that of Full, although it still surpasses the online random selection baseline.

**Different Sample Ratios.** While our main experiments used a sampling ratio of 30%, here we examine how different sampling ratios affect the consistency criterion under both online and offline

Table 3: Online RL results on hard dataset **DAPO-MATH-17K**. Sample ratio is 30%.

| Model | Dataset | Method | GRPO | RLOO | DAPO | REINFORCE++ |
|-------|---------|--------|------|------|------|-------------|
| Qwen2.5-7B | MATH | Random | 68.2 | 72.0 | 70.6 | 71.7 |
| | | Full | **73.2** | **74.8** | **73.5** | 73.9 |
| | | $r_{pb}^{online}$ | 72.9 | 74.3 | 73.2 | **73.9** |
| | GSM8K | Random | 88.1 | 90.1 | 88.5 | 89.4 |
| | | Full | 91.5 | 92.0 | **91.4** | 91.7 |
| | | $r_{pb}^{online}$ | **91.7** | **92.1** | 91.0 | 91.4 |

Table 4: Impact on different RLVR algorithms.

| Model | Dataset | Random | Full | PPL | ENT | $r_{pb}^{online}$ |
|-------|---------|--------|------|-----|-----|-------------------|
| Qwen2.5-7B | DAPO-MATH-17K | 27.1 | **34.0** | 33.0 | 32.1 | 33.8 (+6.7%) |

settings, as shown in Figures 3 (2-c) & (2-d). Our results show that with smaller ratios, such as 10%, the reasoning performance of the model decreases noteblely. For example, compared to Full, performance drops by about 1.8% (MATH) and 1.5% (GSM8K) in the online setting, and by about 3.1% and 2.4% in the offline setting. This indicates that too little training data can seriously impair RL training. On the other hand, increasing the ratio to 50% does not provide a clear improvement and approaches the full data result. These findings highlight the importance of selecting a proper query sampling strategy for RL-based reasoning training.

**Different RLVR Algorithms.** Our main experiments adopt GRPO as the default RLVR algorithm. Here, we further assess whether consistency sampling benefits other RLVR algorithms under the online setting. Results in Table 3 demonstrate that, with a 30% sample ratio, consistency sampling reliably improves the reasoning performance of all tested algorithms, achieving results comparable to Full.

**Hard Training Data.** We also increased the difficulty of the mathematical tasks by validating the importance of consistency sampling on the DAPO-MATH-17K dataset[3], shown in Table 4. Even with a 30% sampling ratio, we observed similar findings as in the main experiments. This confirms that uncertainty consistency query selection is equally effective for more challenging mathematical tasks.

**Selected by Highest Objective Uncertainty.** In the online setting, if we select the top p% samples with the highest objective uncertainty within a batch, these samples are very likely to have rewards of zero under RLVR algorithms. According to the RLVR loss in Equation 1, such samples will contribute no gradient signal, severely hindering model optimization. So we only consider it in offline setting. In the offline setting, we compare two selection strategies on MATH dataset: a) Selecting the top 30% samples with the highest objective uncertainty (Top 30% Hard); b) Selecting 30% samples according to the offline metric $r_{pb}$. The result is shown in Table 5.

Table 5: Training on highest objective uncertainty dataset (Top Hard) vs uncertainty consistency dataset ($r_{pb}$) in the offline setting.

| Model | $r_{pb}$ | Top Hard |
|-------|----------|----------|
| Qwen2.5-7B | **72.1** | 68.3 |
| Qwen2.5-3B | **63.3** | 57.8 |
| Llama3.1-8B-Instruct | **51.5** | 50.4 |

The result demonstrates that filtering samples solely based on objective uncertainty will induce difficulty imbalance in the training set, which can affect the model's generalization ability. Therefore,

---

[3]http://huggingface.co/datasets/BytedTsinghua-SIA/DAPO-Math-17k

our framework explicitly considers uncertainty consistency, rather than relying on a single notion of uncertainty.

## 5.4 DISCUSSION

We further analyzed how consistency sampling affects response length and policy entropy during RL training on the Qwen2.5-7B MATH task, as illustrated in Figures 3 (2-a) & (2-b):

**Response Length.** We find that consistency samples help maintain the model's response length at a level comparable to RL with the full dataset. This suggests the difficulty of the selected samples matches that of the full dataset, while random sampling may pick samples that are too easy and less beneficial for improving reasoning ability.

**Entropy.** With full-data RL, the model's entropy drops rapidly and is noticeably lower than other methods at the early stage of training. This phenomenon limits the model's exploration ability and restricts gains in reasoning ability, even with more data. In contrast, consistency sampling maintains higher entropy in the early phase, promoting better exploration and improving the sample efficiency of RL training.

## 6 CONCLUSION

This work mainly explores what kind of data is more valuable for RLVR training and how to achieve the reasoning effect of full-data RL with less data. Key findings are: 1) Classic active learning strategies underperform full-data RLVR because they ignore objective uncertainty, i.e., the probability that the model answers correctly. 2) We introduce an offline uncertainty consistency metric, which is the point-biserial correlation between correctness and model perplexity; 3) Because of limited sampling and dynamically shifting output distributions, we estimate the online uncertainty consistency metric from normalized advantages and current subjective uncertainty 4) We prove that online metric is strictly negatively correlated with its offline counterpart and maximizing online uncertainty consistency is equivalent to maximizing decrease of sample uncertainty, providing a principled selection criterion. Experiments show that selecting by low uncertainty consistency metric already surpasses random and classic AL baselines, while online selection with only 30% of the data matches or exceeds the accuracy of training on the full dataset. Our consistency-driven query selection thus offers a scalable path to data-efficient RL for complex reasoning tasks.

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

# A APPENDIX

## A.1 PROOF

### A.1.1 PROOF FOR THEOREM 1

**Theorem** 1(Negative Correlation between $r_{pb}$ and $r_{pb}^{\text{online}}$). *For the same model $\pi_\theta$, the covariance between $r_{pb}$ and $r_{pb}^{online}$ is less than zero, i.e., $\text{Cov}(r_{pb}, r_{pb}^{online}) < 0$.*

*Proof.* We first define $r'_{pb} = (\bar{U}_1 - \bar{U}_0)\sqrt{\frac{K_0 K_1}{K^2}}$. Since the variance $s_K > 0$, $\text{Cov}(r_{pb}, r_{pb}^{\text{online}})$ and $\text{Cov}(r'_{pb}, r_{pb}^{\text{online}})$ have the same sign. We now show that $\text{Cov}(r'_{pb}, r_{pb}^{\text{online}}) < 0$. For clarity of exposition, we abstract the above theorem as follows:

> *Let $U$ be a random variable such that $P(U > 1) = 1$, and consider i.i.d $K$ samples $\{u_i\}_{i=1}^K$. Let $P = r'_{pb} = \sum_{i=1}^K c_i u_i$, $Q = r_{pb}^{online} = \sum_{i=1}^K d_i \frac{1}{u_i}$, where $c_i d_i > 0$ for any $i \in [K]$. Then, we have $\text{Cov}(P, Q) < 0$.*

First, expand the expectation of the product:

$$
\mathbb{E}[PQ] = \mathbb{E}\left[\left(\sum_{i=1}^K c_i u_i\right)\left(\sum_{j=1}^K d_j \frac{1}{u_j}\right)\right]
$$
$$
= \sum_{i=1}^K \sum_{j=1}^K c_i d_j \mathbb{E}[u_i \cdot 1/u_j].
$$

For $i = j$, $\mathbb{E}[u_i \cdot 1/u_i] = 1$. For $i \neq j$, if $u_i$ and $u_j$ are independent samples from $U$, then $\mathbb{E}[u_i \cdot 1/u_j] = \mathbb{E}[u_i] \cdot \mathbb{E}[1/u_j]$.

So,

$$
\mathbb{E}[PQ] = \sum_{i=1}^K c_i d_i \cdot 1 + \sum_{i \neq j} c_i d_j \mathbb{E}[u_i]\mathbb{E}[1/u_j]
$$
$$
= \sum_{i=1}^K c_i d_i + \left(\sum_{i=1}^K c_i \mathbb{E}[u_i]\right)\left(\sum_{j=1}^K d_j \mathbb{E}[1/u_j]\right) - \sum_{i=1}^K c_i d_i \mathbb{E}[u_i]\mathbb{E}[1/u_i]
$$

The expectations of $P$ and $Q$ are

$$
\mathbb{E}[P] = \sum_{i=1}^K c_i \mathbb{E}[u_i], \qquad \mathbb{E}[Q] = \sum_{j=1}^K d_j \mathbb{E}[1/u_j].
$$

So,

$$
\mathbb{E}[P]\mathbb{E}[Q] = \left(\sum_{i=1}^K c_i \mathbb{E}[u_i]\right)\left(\sum_{j=1}^K d_j \mathbb{E}[1/u_j]\right)
$$

Subtracting, the covariance becomes:

$$
\text{Cov}(P, Q) = \mathbb{E}[PQ] - \mathbb{E}[P]\mathbb{E}[Q]
$$
$$
= \sum_{i=1}^K c_i d_i - \sum_{i=1}^K c_i d_i \mathbb{E}[u_i]\mathbb{E}[1/u_i]
$$

So,

$$
\text{Cov}(P, Q) = \sum_{i=1}^K c_i d_i \left(1 - \mathbb{E}[u_i]\mathbb{E}[1/u_i]\right)
$$

Now, because $u_i > 1$ almost surely, we know $\mathbb{E}[u_i] > 1$ and $\mathbb{E}[1/u_i] < 1$. More importantly, by Jensen's inequality or the Cauchy-Schwarz inequality,

$$\mathbb{E}[u_i]\mathbb{E}[1/u_i] \geq 1$$

with equality only if $u_i$ is constant. Therefore,

$$1 - \mathbb{E}[u_i]\mathbb{E}[1/u_i] < 0$$

and, since $\sum_{i=1}^{K} c_i d_i > 0$, it follows that

$$\text{Cov}(P, Q) < 0.$$

So, $\text{Cov}(r_{pb}, r_{pb}^{online}) < 0$.

$\square$

### A.1.2 PROOF FOR THEOREM 2

**Theorem** 2 (Equivalent between Maximizing Decrease in Sample Uncertainty and Maximizing $r_{pb}^{online}$). *Suppose $U(x; \theta) = \sum_{j=1}^{K} U_j^\theta$ denotes the subjective uncertainty for sample $x$. Under sample gradient orthogonality assumption (Assumption 1) and bounded gradient norm assumption (Assumption 2), in one optimization step of an on-policy RL algorithm (e.g., GRPO), selecting samples in the minibatch with largest $r_{pb}^{online}$ can maximize the decrease in sample uncertainty .*

First, assumption 2 is common and mild. Now we explain why assumption 1 is reasonable in practice. We use Qwen2.5-0.5B to perform eight inference runs on the same question. For each response, we compute the sample derivative of the uncertainty with respect to the parameters and the normalized gradient inner products for each pair of derivative. As Figure 4 shown, apart from the diagonal, the gradient inner products between different responses are close to zero. This indicates that, although the samples are generated for the same question, the uncertainty gradients in the high-dimensional parameter space remain approximately orthogonal. Next, we begin to prove Theorem 2.

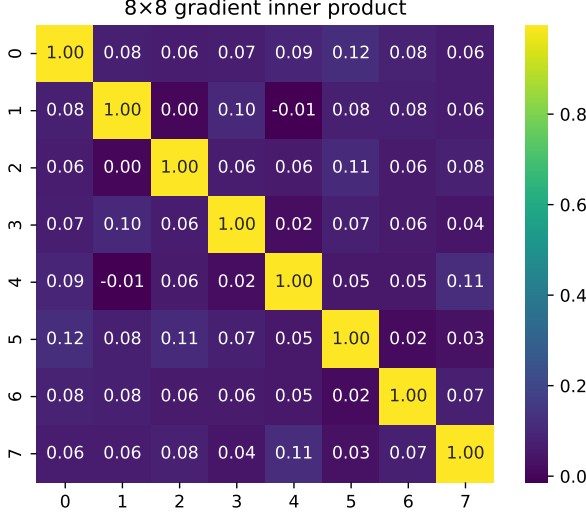

Figure 4: Gradient inner product heatmap.

*Proof.* For any query $x$, we have $K$ responses $\{y_i\}_{i=1}^K$ and set $\gamma = \frac{M^2}{m^2}$. And according to on-policy RL algorithm (e.g., GRPO), the loss is defined as follows:

$$\mathcal{L}^\theta = -\frac{1}{K}\sum_{i=1}^K \hat{A}_i \frac{1}{|y_i|}\sum_{t=1}^{|y_i|}\log\pi_\theta(y_t|x,y_{<t})$$

$$= \frac{1}{K}\sum_{i=1}^K \hat{A}_i \ln e^{-\frac{1}{|y_i|}\sum_{t=1}^{|y_i|}\log\pi_\theta(y_t|x,y_{<t})}$$

$$= \frac{1}{K}\sum_{i=1}^K \hat{A}_i \ln U_i^\theta. \qquad\qquad \text{Equation 3}$$

Taking the derivative of the above equation with respect to $\theta$, we have:

$$\nabla_\theta\mathcal{L}^\theta = \frac{1}{K}\sum_{i=1}^K \frac{\hat{A}_i}{U_i^\theta}\nabla_\theta U_i^\theta.$$

According to Gradient Decent, the parameters in the next step is:

$$\theta' = \theta - \eta\nabla_\theta\mathcal{L}^\theta,$$

where $\eta$ is learning rate. the decrease in sample uncertainty after this update is:

$$\Delta U(x) = U(x;\theta') - U(x;\theta)$$

$$\approx -\eta\nabla_\theta U(x)^T\nabla_\theta\mathcal{L}^\theta \qquad\qquad \text{First-order Taylor Estimation}$$

$$= -\eta\left(\sum_{i=1}^K \nabla_\theta U_i^\theta\right)\left(\frac{1}{K}\sum_{i=1}^K \frac{\hat{A}_i}{U_i^\theta}\nabla_\theta U_i^\theta\right)$$

$$= -\frac{\eta}{K}\sum_{i=1}^K \frac{\hat{A}_i}{U_i^\theta}\sum_{j=1}^K <\nabla_\theta U_i^\theta, \nabla_\theta U_j^\theta>$$

$$= -\frac{\eta}{K}\sum_{i=1}^K \frac{\hat{A}_i}{U_i^\theta}||\nabla_\theta U_i^\theta||_2^2 \qquad\qquad \text{Assumption 1}$$

$$= -\frac{\eta}{K}\left(\sum_{\hat{A}_i>0} \frac{\hat{A}_i}{U_i^\theta}||\nabla_\theta U_i^\theta||_2^2 + \sum_{\hat{A}_i<0} \frac{\hat{A}_i}{U_i^\theta}||\nabla_\theta U_i^\theta||_2^2\right)$$

$$\leq -\frac{\eta}{K}\left(m^2\sum_{\hat{A}_i>0} \frac{\hat{A}_i}{U_i^\theta} + M^2\sum_{\hat{A}_i<0} \frac{\hat{A}_i}{U_i^\theta}\right) \qquad\qquad \text{Assumption 2}$$

$$= -\frac{\eta m^2}{K}\left(\sum_{\hat{A}_i>0} \frac{\hat{A}_i}{U_i^\theta} + \gamma\sum_{\hat{A}_i<0} \frac{\hat{A}_i}{U_i^\theta}\right)$$

$$= -\eta m^2 r_{pb}^{online}$$

So, maximize the decrease in sample uncertainty is equivalent to maximize $r_{pb}^{online}$. $\qquad\qquad \square$

## B  MARGIN SCORE AND ENTROPY ESTIMATION

Similarly, we can use the Margin Score (MS) or Entropy (ENT) (Wang & Shang, 2014) to represent the subjective uncertainty of LLMs:

$$MS_k^{(i)} = \frac{1}{|y_k^{(i)}|}\sum_{t=0}^{|y_k^{(i)}|}\left(\pi_{ref}(y_{m,t}|x^{(i)}, y_{k,<t}^{(i)}) - \pi_{ref}(y_{s,t}|x^{(i)}, y_{k,<t}^{(i)})\right), \qquad (6)$$

$$ENT_k^{(i)} = \frac{1}{|y_k^{(i)}|} \sum_{t=0}^{|y_k^{(i)}|} \mathcal{H}(\cdot | x^{(i)} y_{k,<t}^{(i)}). \tag{7}$$

Here, $y_{m,t}$ and $y_{s,t}$ denote the tokens with the highest and second-highest probabilities at position $t$ in the current sequence, respectively. $\mathcal{H}(\cdot | x, y)$ represents the entropy at each position of the current sequence. A larger MS or a smaller ENT reflects greater subjective uncertainty of the LLMs.

## C  TASK PROMPT

---
**MATH Task Prompt**

[User]
<Question>
Let's think step by step and output the final answer within \\boxed{}.

---
**GSM8K Task Prompt**

[User]
<Question>
Let's think step by step and output the final answer after ####.

---
**AskLLM (Sachdeva et al., 2024) Prompt**

###
Question
###

Does the previous reasoning question demarcated within ### and ### contain informative signal for reasoning reinforcement learning training ? An informative datapoint should be well-formatted, contain some usable knowledge of the world, and strictly NOT have any harmful, racist, sexist, etc. content. This reasoning question should have a clear answer, and you should consider it solvable for you, while also ensuring that it is not an overly simple question.

OPTIONS:
- yes
- no

---

## D  WARM UP EXPERIMENT SETUP

In Section 1, we assess classic AL strategies in the offline reasoning RL setting. We check Full , Random , uncertainty-based methods (PPL, Entropy (Wang & Shang, 2014)), feature-sapce coverage methods (K-means, K-center (Sener & Savarese, 2017)) and LLM prompting base methods (AskLLM (Sachdeva et al., 2024)) using GRPO on Qwen2.5-0.5B (Qwen et al., 2025) in MATH (Hendrycks et al., 2021) dataset. The detailed experimental setup is as follows:

- **Full:** Training is conducted on the entire 7,500 samples from the MATH dataset.

- **Random:** 10% of the training data are randomly sampled for training.

- **PPL:** Training samples are selected as the top 10% of the training data with the highest average perplexity (Equation 3).

- **Entropy:** Training samples are selected as the top 10% of the training data with the highest average entropy (Equation 7).

- **K-means:** After obtaining vector representations from the Qwen3-8B-Embedding (Zhang et al., 2025) model for each query, k-means++ (Arthur & Vassilvitskii, 2006) initialization and k-means clustering are applied, and 10% of the samples are uniformly drawn from each cluster for training.

- **K-center:** After obtaining vector representations from the Qwen3-8B-Embedding model (Zhang et al., 2025), the K-center (Sener & Savarese, 2017) algorithm is applied to select the top 10% of samples for training.

- **AskLLM:** Using the AskLLM prompt (See Appendix C), the probability of the token "yes" appearing in the model's response is recorded, and the top 10% of samples with the highest "yes" probability are selected for training.

All the hyperparameters in the warm up experiment is the same as those in main experiment (See Section 5.1. Besides, every experiment is conduct 5 times using different random seed and we report its mean and standard deviation of greedy accuracy on 5000 MATH test dataset.

## E    THE USE OF LARGE LANGUAGE MODELS (LLMS)

The Abstract, Introduction, and Method sections of the paper are polished with the assistance of a large language model.

