# OpenReview forum: "Learn More with Less: Uncertainty Consistency Guided Query Selection for RLVR"
_ICLR.cc/2026/Conference — ICLR 2026 Poster_

### Official Review · Reviewer_KvRw · 2025-10-23

**Soundness:** 3
**Presentation:** 4
**Contribution:** 4
**Rating:** 8
**Confidence:** 4

**Summary:**

The authors are proposing to use Active Learning (AL) to mitigate the large sample requirements of Reinforcement Learning with Verifiable Reward (RLVR) used in the LLM alignment process.
They find, that samples where the predicted model uncertainty diverges from the actual accuracy on the sample (measured by sampling K outputs and computing the reward) are detremental to model convergence in RLVR.
The authors propose an AL score to find samples with high alignment of uncertainty and accuracy and use this score to sample a subset of the full training data (offline setting).
Since this proposed score is difficult to obtain in the online setting, the authors also propose an approximation of their score and demonstrate its theoretical properties.
Both proposed methods show strong empirical results on 3 models and 2 math datasets.

**Strengths:**

- Very streamlined description of research gap, related work and proposed method
- Excellent line of reasoning:
	1. Preliminary experiment to show shortcomings of existing solutions to the problem
	2. Starting with a simplyfied setting (offline RL) and showing a principled advantage of the proposed method
	3. Generalizing to realistic settings (online RL) by finding an approximation of their method that has theoretical guarantees
	4. Demonstrating strong performance of the approximation on 3 models and 2 datasets
	5. Providing ablation studies on the additional properties of their method
- Providing believable evidence for the non-standard assumption of "Sample Gradient Orthogonality"
- No additional computational overhead, as sampling K inferences for each x is already part of RLVR

**Weaknesses:**

- the authors claim "(non-)significant lifts" multiple times (line 78,88,373,381), but do not provide hard evidence for this claim in the form of standard deviations of results, critical difference diagrams or p-tests. We urge the authors to either provide one of these metrics in the appendix, or use a less mathematically loaded term instead of "significant".

**Questions:**

- Impact of higher gammas (ablation study): Higher gamma values mean a stronger deterrence of predicted negative advantage in Eq. 5. Does this mean, we focus on samples the model knows to improve upon (positive advantage for some output y)? If so, does this not direcetly counteract exploration behaviours in the RL training? Would this be comparable to directly influencing exploration/exploitation ratios in the training?

---

> ### Author Response · Authors · 2025-11-17
> **Response to Reviewer KvRw**
>
> We sincerely appreciate your thorough and insightful review of our work, as well as your strong recognition of its advantages in terms of **writing**, **motivation**, **scenario formulation**, **empirical performance**, **verification of assumption validity,** and **computational efficiency**. Below, we provide clarifications and discussions in response to the valuable weaknesses and questions you raised.
>
> **Response to Weakness**:
>
> We apologize for our misuse of the term “significant” without performing statistical significance testing. In **Table 1**, each experiment was repeated five times, and we reported the corresponding mean and variance. However, in the Main Results section, due to constraints in time and computational resources, it was challenging to run extensive repetitions and compute appropriate statistical significance measures. In the revised manuscript, we will replace “significant” with more gentle word  to avoid misleading implications.
>
> ---
>
> **Response to Question**:
>
> Your perspective is both **novel and thought-provoking**. Indeed, as shown in our ablation results, a high $ \gamma $ online metric can negatively impact both the quality of query selection and the model’s generalization ability. This highlights the importance of focusing more on consistency metrics for positive samples. For a positive sample, the model should output its trajectory with high probability; after the model update, the output probability for that sample is reinforced. At first glance, this process may appear to harm the model’s exploration capability. However, as illustrated in **Fig. 2(b)**, query selection using $ r_{pb}^{online} $ produces a higher entropy in convergence compared with fine-tuning on the full dataset — indicating, from an empirical standpoint, an increase in the model’s exploration capability.
>
> We believe the reason behind this apparent contradiction **lies in the role of high-model-uncertainty inconsistent positive samples**. Such samples tend to produce large gradient noise. When these samples are included in backpropagation, they can actually constrain exploration more severely and impair the RL training process.
> Finally, regarding your observation that $ \gamma $ influences the exploration/exploitation ratio: both our theoretical proof and experimental results support this as a reasonable explanation. We sincerely thank you for providing this insightful perspective, which has helped reveal some fundamental aspects of our method.

---

### Official Review · Reviewer_4djn · 2025-10-31

**Soundness:** 2
**Presentation:** 3
**Contribution:** 3
**Rating:** 4
**Confidence:** 4

**Summary:**

This work introduces the active learning (AL) process with a new acquisition metric into reinforcement learning with verifiable reward (RLVR) for large language models (LLMs). They first confirm the query selection's impact on the stability of the gradients during fine-tuning via RLVL and reveal that the classic AL cannot select the most informative queries as well as random sampling. After assessing the inconsistent between an LLM's output with the lowest probability and a reward model's evaluation of the samples' accuracy effects on the gradient norm, they propose the smallest inconsistent metric $r_{pb}$ by the Point-Biserial Correlation Coefficient (PBC). Moreover, they extend the idea to online training. The experimental results show that the AL with an inconsistent metric could enhance both offline and online training as well as different RL algorithms.

**Strengths:**

1. The story (writing) is good to easily follow the authors' idea and refresh the utilization of the AL for the emerging field.
2. This paper highlights that the importance of the query selection metric should not only rely on the LLM itself but also require consistency with the reward model's evaluation.

**Weaknesses:**

1. **My concern about using uncertainty.** After reviewing the Eqs. (2) and (3), IIUC, the definition of the subjective uncertainty is the low average probability of a policy model's responses $\log \pi_\mathrm{ref}(y_{k, t}^{(i)} \vert x^{(i)}, y_{k, <t}^{(i)})$ and the objective uncertainty is the low accuracy of a reward model's evaluation of a model's response, respectively. However, why can we call these two terms uncertainty? For example, if an LLM's response gives a response with lower probability but a 'consistent response' for the same (or similar) prompt $x^{(i)}$, could we still say the sample is uncertain?
2. Follow 1., I feel that the metric of these terms is more like 'difficulty of reasoning the sample $x^{(i)}$', i.e., the degree of the probability that LLM can give the response (reasoning) and get the high reward (correct answer) for a sample. If so, what you check is the consistency between an LLM's output and the reward model's output.
3. **My concern about the comparison with other AL methods.** Follow 2, if the core idea is evaluating a degree of the informative sample requires both LLMs and reward models, the proposed comparison with Entropy, K-center, K-means, and AskLLM might be insufficient, which only considers the LLMs' response and ignores the reward models' evaluation. To strengthen it, I suggest that the authors consider adding an alternative ablation study on the 'uncertainty' of the reward models.
4. **Make a consistent symbol for equations.** For example, Eq. (1) uses $y_i$ but Eq. (3) uses $y_k^{(i)}$, the index of the sample and the index of generations should be differentiated.
5. In Figure 1, the authors present that the inconsistent sample would give high gradient norm dynamics, i.e., these samples would cause gradient instability. However, the (degree of) impact of these gradient instabilities on the final performance is unclear. To highlight the gradient instability would be a significant issue for RLVR, I suggest that the authors provide some illustrations or examples of this issue.

**Questions:**

1. While you mention that *Because the calculation of $r_{pb}$ relies on a large number of samples $K$, ...* in Sec 4.2, your experimental settings of $K = 8$ in Sec 5.1 seems not large. Could you give stronger motivations for using Online Query Selection? For example, the **Model update** is the key point to address.
2. Following 1., what are the key components in Online Query Selection for addressing sampling distribution shift in **Model update**?

---

> ### Author Response · Authors · 2025-11-17
> **Response to Reviewer 4djn**
>
> We sincerely appreciate Reviewer 4djn’s positive evaluation of the **writing** and **motivation** of our work. However, we are sorry that some ambiguities in our presentation may have led to misunderstandings regarding the contributions of our study. Below, we provide the following clarifications to each of the weaknesses you raised.
>
> **Response to Weakness 1 & 2**:
>
> Your understanding of our definitions of subjective uncertainty and objective uncertainty is completely correct. However, regarding your comment: “_if an LLM’s response gives a response with lower probability but a 'consistent response' for the same (or similar) prompt, could we still say the sample is uncertain?_”  Indeed, such a sample can be regarded as subjectively uncertain, since the entire response trajectory is generated with low probability. If this low-probability response is nevertheless a consistent response in our sense, it means that the final output is incorrect — implying the sample is also objectively uncertain. Based on both our theoretical analysis and empirical results, such consistent responses are actually beneficial for RL training of the model.
> We suspect that there may be a misunderstanding regarding our definition of consistent response. To clarify: **in our context, consistent response refers to cases where (i) low-probability generations are likely to be wrong, or (ii) high-probability generations are likely to be correct — rather than producing similar answers for the same prompt.**
>
> Furthermore, you correctly pointed out that these two metrics can describe the difficulty of a sample $ x^{(i)} $:
>
> * High subjective uncertainty indicates that the model encountered an “out-of-distribution” sample, prompting it to sample from a low-probability region.
> * High objective uncertainty indicates that no matter whether the model samples from low- or high-probability regions, it is very likely to fail on the problem.
>
> Therefore, filtering samples solely based on subjective or objective uncertainty **will induce difficulty imbalance in the training set (too hard or too easy)**, which can predictably affect the model’s generalization ability. Our framework therefore explicitly considers consistency as discussed above, rather than relying on a single notion of uncertainty.
>
> ---
>
> **Response to Weakness 3**:
>
> In the online setting, if we select the top-(p)% samples with the highest objective uncertainty within a batch, these samples are very likely to have rewards of zero under RLVR algorithms (such as GRPO). According to the RLVR loss in **Eq. 1**, **such samples will contribute no gradient signal**, severely hindering model optimization.
>
> In the offline setting, we compared two selection strategies on MATH dataset:
> * selecting the top 30% samples with the highest objective uncertainty (**Top 30% Hard**)
> * selecting 30% samples according to the offline metric $ r_{pb} $.
>
> Experimental results is shown as follow:
>
> | Model             | $ r_{pb} $ | Top Hard |
> |-------------------|----------|----------|
> | Qwen2.5-7B        | **72.1**     | 68.3     |
> | Qwen2.5-3B        | **63.3**     | 57.8     |
> | Llama3.1-8B-Instruct | **51.5**  | 50.4     |
>
>
> The results confirm our earlier analysis: naively selecting **“hard”** samples induces difficulty imbalance in training set, which substantially affects the model’s generalization performance.
>
> ---
>
> **Response to Weakness 4 and Typo error**:
>
> We thank you for your careful attention to detail. We have corrected and unified the relevant typo errors in the manuscript.
>
> ---
>
> **Response to Weakness 5**:
>
> In the conventional **REINFORCE** algorithm for reinforcement learning, directly using the Q-function as the scaling term for the policy gradient estimation often results in high variance. To reduce the variance of this gradient estimate, it is common practice to subtract a baseline $ b(s) $ (typically implemented as a state value function $ V(s) $) from the Q-function. This approach preserves the unbiasedness of the estimator while lowering the variance of the single-step gradient estimation, thereby improving the stability of model training. **This observation highlights that, in reinforcement learning, mitigating the variance in the gradient norm is crucial for ensuring stable training.**
>
> Besides, a direct theoretical analysis in this context is challenging, since the gradient norm’s variation depends on several intertwined factors:
> * the order in which samples are fed to the model,
> * optimizer hyperparameters
> * the precise mathematical definition of performance (i.e., generalization).

---

> > ### Author Response · Authors · 2025-11-17
> > **Response to Reviewer 4djn (2)**
> >
> > **Response to Question**:
> >
> > We apologize for the misuse of the symbol ($ K $) in **Sec. 4.2**. On line 312, we stated that in the offline setting, each sample is drawn 64 times to accurately estimate the offline metric $ r_{pb} $. In the online setting, each sample is drawn only 8 times to estimate the online metric $ r_{pb}^{online} $. We regret the confusion caused by using the same symbol $ K $ for both settings.
> > Additionally, to address the issue of sampling distribution shift, we replace the reference model $ \pi_{\text{ref}} $ with the training model $\pi_\theta $ during the RL process. Specifically, in Eq. 4, the variable $ U $ is estimated using $ \pi_{\text{ref}} $, whereas in **Eq. 5** it is estimated using $ \pi_\theta $. This mitigates the sampling distribution shift introduced by parameter updates.

---

### Official Review · Reviewer_ntf7 · 2025-11-01

**Soundness:** 3
**Presentation:** 3
**Contribution:** 3
**Rating:** 6
**Confidence:** 3

**Summary:**

This paper investigates query selection strategies for reinforcement learning from vision and reward (RLVR). The authors observe that standard active learning (AL) sampling methods often fail to outperform random selection in this context. To address this, they introduce an uncertainty consistency metric to guide sampling. In the offline setting, they use PBC (policy–behavior consistency) to measure alignment, while for online training—where estimating PBC directly is difficult—they propose a variant based on normalized advantage and subjective uncertainty. The paper also provides a theoretical analysis suggesting a negative correlation between offline and online PBC. Empirically, the proposed approach outperforms both random and classic AL baselines, reaching near–full-dataset performance using only 30% of the data.

**Strengths:**

1. The problem is well-motivated and relevant to current challenges in RLVR.
2. The paper offers an interesting empirical observation that inconsistent samples can lead to extreme gradients, which explains why standard AL can underperform random sampling.
3. The introduction of two alignment metrics—one for offline and one for online settings—is insightful, and the accompanying theoretical analysis provides some grounding.
4. Experiments are extensive and demonstrate strong results, achieving competitive performance with significantly fewer samples.

**Weaknesses:**

While the paper is promising, several points could benefit from deeper clarification or justification:

1. The link between sample inconsistency and extreme gradient behavior is intuitively explained but lacks theoretical support or formal analysis.
2. It is unclear why the offline setting cannot also leverage the online metric $r_{pb}^{online}$, which appears to yield stronger performance in experiments.
3. In some cases, training on the full dataset leads to worse results than using only 30% of the data; the paper should provide more discussion or intuition for why this happens.

Typo: In line 451, “Table ??” is not rendering correctly.

**Questions:**

Please comment/justify the weakness part above.

---

> ### Author Response · Authors · 2025-11-17
> **Response to Reviewer ntf7**
>
> We sincerely appreciate Reviewer ntf7’s thorough recognition of the **motivation**, **experimental design**, **theoretical guarantees**, and **empirical performance** of our work.
> Regarding the three insightful weaknesses you raised, we provide the following clarifications and explanations:
>
>
> **Response to To Weakness 1**:
>
> To be honest, a direct theoretical analysis in this context is challenging, since the gradient norm’s variation depends on several intertwined factors:
> * the order in which samples are fed to the model
> * optimizer hyperparameters
> * the precise mathematical definition of performance (i.e., generalization).
>
> However, we can intuitively explain why “inconsistent samples” tend to induce larger variance in gradient norms. For positive samples that are inconsistent—i.e., cases where the model assigns relatively low probability to the correct answer—the corresponding trajectory contains many low‑probability tokens such as “think,” “wait,” or “probably.” According to the expression for the loss gradient $\nabla_{\theta}\mathcal{L}^{\theta}$ derived in the **appendix A.1.2**, such samples are inclined to produce gradients with larger norms.
>
> For a concrete illustration, consider that in a consistent positive sample, the average generation probability of tokens might be around 0.9, whereas in an inconsistent positive sample it may be around 0.3. Since policy gradient methods tend to increase the probabilities of positive sample tokens toward 1, inconsistent samples have a larger margin for probability improvement, thereby making large‑norm and unstable gradients more likely. The same reasoning applies to negative samples.
>
> Thus, we obtain an intuitive explanation for why “inconsistent samples” lead to higher variance in gradient norms.
>
> ---
>
> **Response to Weakness 2**:
>
> You mentioned: “_why the offline setting cannot also leverage the online metric , which appears to yield stronger performance in experiments._”  We note several reasons why, under the offline setting, the offline metric (**Eq. 4**) is preferable to the online metric (**Eq. 5**):
>
> * **Direct consistency modeling**: **Eq. 4** explicitly models the consistency between the model’s uncertainty (a continuous variable) and the objective uncertainty (a binary variable). In contrast, **Eq. 5** is merely highly linearly correlated with **Eq. 4**. As shown in **Fig. 1(b)**, when $ r_{pb} $ is relatively large or small, $ r_{pb}^{online} $ exhibits considerable variance. We argue that this variance originates from the fact that  $r_{pb}^{online}$ does not take into account the normalization effect of  $s_k$ within $r_{pb}$.
>
> * **Hyperparameter sensitivity**: The online metric $r_{pb}^{online}$ involves a hyperparameter $\gamma$ which cannot be directly determined from theoretical analysis and requires multiple trials to identify an optimal value. This results in a prohibitively high cost when applied in offline setting.
>
> Therefore, in the offline setting, we choose not to use the online metric.
>
> ---
>
> **Response to Weakness 3**:
>
> As illustrated in **Fig. 1(a)** and supported by **Theorem 2**, for inconsistent samples, both the gradient norm and the single step change of the model’s uncertainty exhibit high variance. Within the full dataset, if such inconsistent samples are not filtered out, the gradients they produce can adversely affect model optimization. Conversely, if we identify inconsistent samples beforehand and exclude them from backpropagation, this can reduce the variance of the gradient norm, enhance the single-step reduction in model uncertainty, and thereby improve both the optimization trajectory and the final generalization performance.

---

### Official Review · Reviewer_eynX · 2025-11-01

**Soundness:** 3
**Presentation:** 3
**Contribution:** 2
**Rating:** 6
**Confidence:** 4

**Summary:**

The paper introduces a new query selection strategy for Reinforcement Learning with Verifiable Rewards (RLVR) that allows training mathematical reasoning models using far fewer queries, without sacrificing performance. The key insight from the paper is that not all queries are equally informative. Standard active learning methods often pick samples with high subjective uncertainty (e.g., high perplexity), but this fails in RL reasoning because these samples frequently produce unstable or high-variance gradients, which hurts training stability. To address this problem, the paper proposes selecting samples where Subjective uncertainty (model confidence) and Objective uncertainty (whether the answer is correct) are consistent. They define it as “uncertainty-consistent” samples. The Experimental results show that using only 30% of the training data, the model reaches the same or better performance on the reasoning tasks compared to the full datasets RLVR training.

**Strengths:**

The paper identifies a practical limitation in current RL-based reasoning training pipelines: query selection methods that rely solely on subjective uncertainty (e.g., perplexity) often select examples that are uncertain but uninformative, leading to unstable gradients and inefficient learning. This motivation is clearly articulated and supported by empirical evidence. This insight is both intuitive and impactful—valuable training samples are those where uncertainty meaningfully reflects correctness, rather than those that are merely hard. The theoretical analysis establishing their negative correlation and training benefit is clearly developed and strengthens the contribution.
The method achieves full-dataset performance using only ~30% of the training data, while maintaining or improving generalization on standard math reasoning benchmarks. This is a practically meaningful result, especially given the rising cost of RL-based reasoning training.

**Weaknesses:**

The consistency metric assumes that the examples used for selection reflect the distributions during RL optimization. If the underlying data distribution shifts over training (which is common in RLVR), the effectiveness of selection may degrade unless the scoring is frequently recomputed.

The evaluation is rather limited to only Math reasoning. For query selection methods, it would be great to draw broader insights on whether the methods can be generalized beyond Math reasoning tasks.

**Questions:**

Do the uncertainty consistency proposed in this paper still hold in open-ended, multi-step reasoning tasks where correctness is subjective or less binary (e.g., instruction following, safety, dialogue)?

---

> ### Author Response · Authors · 2025-11-17
> **Response to Reviewer eynX**
>
> We sincerely appreciate Reviewer eynX’s comprehensive recognition of the **motivation**, **theoretical framework**, **methodology**, and **experimental results** of our work. The two weaknesses and the question you raised are highly specific and constructive. We provide the following clarifications:
>
> **Response to Weakness 1**:
>
> You pointed out that “_The consistency metric assumes that the examples used for selection reflect the distributions during RL optimization. If the underlying data distribution shifts over training , the effectiveness of selection may degrade unless the scoring is frequently recomputed._” This observation is accurate and aligns with our design intent. In fact, this work proposes two different experimental settings—offline setting (**Eq. 4**) and online setting (**Eq. 5**).
>
> In the offline setting, a reference model is used to estimate model uncertainty. Since the estimated data distribution changes continuously throughout RL training, the effectiveness of sample selection based on the offline metric inevitably degrades as training progresses. To address the issue of distributional shift during training, we introduce the online metric ($r_{pb}^{\text{online}}$) (**Eq. 5**). This metric can efficiently compute, with only a small number of samples ($K=8$ in paper), a measure that exhibits a high linear correlation with the offline metric ($r_{pb}$), enabling dynamic sample selection within a batch during RL training. Specifically, within each batch,  $r_{pb}^{\text{online}}$ is computed, and the top p% of samples are selected for backpropagation.
>
> ---
>
> **Response to Weakness 2 and Question**:
>
> You asked whether the proposed method “_it would be great to draw broader insights on whether the methods can be generalized beyond Math reasoning tasks._” For open-ended tasks (e.g., writing, safety, dialogue), we believe the computation of model uncertainty can remain unchanged. However, the computation of rewards differs substantially. In mathematical reasoning tasks, the existence of definitive ground-truth answers allows the final reward variable ( $R$ ) to be modeled as a Bernoulli-distributed binary variable, with negligible noise (i.e., minimal error introduced by answer matching). In contrast, for open-ended tasks, rewards generated by a Scale Reward Model or Generative Reward Model are continuous variables or follow multinomial distributions, **rendering the computation of the consistency metric (**Eqs. 4 & 5**) more complex, and existing much more noise to get the reward signal**. Therefore, the proposed method can be directly transferred and applied to all tasks with binary rewards. For tasks with continuous or multinomial rewards, further adaptation of the consistency metric would be required.

---

### Author Response · Authors · 2025-12-03
**Summary**

We sincerely thank the reviewers for their thorough, insightful, and constructive feedback, and we are grateful to the Area Chair for overseeing this process. The reviewers' comments have been invaluable in helping us clarify our contributions and significantly strengthen the paper.

To summarize our detailed responses, our key clarifications and enhancements include:

*   **Empirically-Backed Justification for "Consistency":** We went beyond theoretical arguments by providing **new experimental data**. This evidence directly validates our core claim: our "consistency" filtering significantly outperforms naive "hard sample" mining, which we show can unbalance the training set and harm generalization. This addresses a central question about our method's efficacy.

*   **Strategic Design of Offline vs. Online Metrics:** We clarified that the offline/online split is a **deliberate design choice** to address the exact problem of distribution shift raised by  Reviewer 4djn. The offline metric establishes the principle, while the efficient online metric is specifically designed to adapt dynamically during RL training, turning a perceived weakness into a core strength of our methodology.

*   **Collaborative and Insightful Engagement:** We embraced the reviewers' suggestions to deepen the paper's insights. For instance, we adopted  Reviewer KvRw's novel perspective on the **exploration-exploitation trade-off**, using it to provide a richer theoretical explanation for our empirical results (i.e., higher entropy). This demonstrates our active engagement with the review process.

*   **Connecting to Foundational RL Principles:** We strengthened our motivation by explicitly connecting our method to the established RL principle of **gradient variance reduction** (analogous to baselines in REINFORCE). This frames our work within a well-understood context, making the intuition for why filtering "inconsistent" samples stabilizes training much clearer.

We believe these clarifications, new results, and deeper theoretical connections have substantially improved the manuscript. We are grateful for the opportunity to have this dialogue and look forward to any further guidance.

---

### Meta-Review · Area_Chair_8GyH · 2026-01-09

**Summary:**

The authors propose an active learning selection algorithm for RLVR.  It introduces an uncertainty-consistency criterion that favors samples where subjective uncertainty (model confidence) aligns with objective uncertainty (verifiable correctness), using an offline PBC-based metric and an online approximation based on normalized advantage.  Empirically it matches full-dataset RLVR performance using only ~30% of the data and the authors also conduct preliminary theoretical study.

We note that using consistency in active learning is NOT new. There are difference consistency design for different task in ML. To name a few:

Consistency-Based Semi-Supervised Active Learning: Towards Minimizing Labeling Cost, ECCV 2020

There are many more, please have a short discussion of those works in related works.

**Reviewer Concerns:**

Addressed: Distribution shift, online setting for offline, hard sample mining baseline, naming discussion

Outstanding: beyond binary, formal theory study

**Reviewer Scores:**

Reviewer 4djn might raise 1 point as the main concern is around naming.

---

### Decision · Program_Chairs · 2026-01-26

Accept (Poster)